# Identifying Clusters of Adherence to Cardiovascular Risk Reduction Behaviors and Persistence with Medication in New Lipid-Lowering Drug Users. Impact on Healthcare Utilization

**DOI:** 10.3390/nu13030723

**Published:** 2021-02-25

**Authors:** Sara Malo, María José Rabanaque, Lina Maldonado, Belén Moreno-Franco, Armando Chaure-Pardos, María Jesús Lallana, María Pilar Rodrigo, Isabel Aguilar-Palacio

**Affiliations:** 1Department of Preventive Medicine and Public Health, University of Zaragoza, 50009 Zaragoza, Spain; rabanake@unizar.es (M.J.R.); mbmoreno@posta.unizar.es (B.M.-F.); achaure@salud.aragon.es (A.C.-P.); iaguilar@unizar.es (I.A.-P.); 2Fundación Instituto de Investigación Sanitaria de Aragón (IIS Aragón), 50009 Zaragoza, Spain; lmguaje@unizar.es (L.M.); mjlallana@salud.aragon.es (M.J.L.); 3Grupo de Investigación en Servicios Sanitarios de Aragón (GRISSA), Spain; mrodrigo@aragon.es; 4Department of Structure and Economic History and Public Economy, University of Zaragoza, 50009 Zaragoza, Spain; 5Centro de Investigación Biomédica en Red Enfermedades Cardiovasculares (CIBERCV), Spain; 6Aragon Healthcare Service, 50017 Zaragoza, Spain; 7Aragon Department of Health, 50017 Zaragoza, Spain

**Keywords:** cardiovascular disease, primary prevention, medication adherence, healthy lifestyle, workplace

## Abstract

We sought to identify specific profiles of new lipid-lowering drug users based on adherence to a healthy lifestyle and persistence with medication, and to characterize co-morbidities, co-treatments, and healthcare utilization for each of the profiles identified. Observational study in 517 participants in the Aragon Workers’ Health Study (AWHS) without previous cardiovascular disease (CVD) and who initiated lipid-lowering therapy. Data were collected from workplace medical examinations and administrative health databases (2010–2018). Using cluster analysis, we identified distinct patient profiles based on persistence with therapy and lifestyle. We then compared characteristics, morbidity, and healthcare utilization across clusters. Participants were aggregated into four clusters based on persistence with therapy, smoking status, adherence to Mediterranean diet, and physical activity. In cluster 1 (*n* = 113), comprising those with a healthiest lifestyle (14.2% smokers, 84.0% with medium-high adherence to Mediterranean diet, high physical activity), 16.8% were persistent. In cluster 3 (*n* = 108), comprising patients with the least healthy lifestyle (100% smokers, poor adherence to the Mediterranean diet, low level of physical activity), all were non-persistent. Clusters 2 (*n* = 150) and 4 (*n* = 146) both comprised patients with intermediate lifestyle behaviors, but differed in terms of persistence (100 and 0%, respectively). Compared with other clusters, the burden of morbidity, cardiovascular score, and healthcare utilization were lower in cluster 1. The healthy adherer effect was only observed in new lipid-lowering drug users of certain profiles. Furthermore, we found that differences in adherence to lifestyle and medication recommendations for CVD prevention influenced morbidity burden and healthcare utilization.

## 1. Introduction

Cardiovascular disease (CVD) is the leading cause of death in Europe [1]. Individual control of CVD risk factors is based on health-promoting behaviors, consisting primarily of smoking cessation, a healthy diet, regular exercise, weight control, and reduced stress. When lifestyle changes are insufficient, it is advisable to establish evidence-based preventive therapy [2]. The effectiveness of CVD preventive drugs is strongly associated with medication-use behaviour [3,4,5].

The “healthy adherer effect”, whereby individuals who adhere to medication tend to also adhere to healthy lifestyle behaviors, and thus have a better health status, is well described [6]. Indeed, this effect is considered a common source of bias in observational studies [7]. Studies have shown that statin users with acceptable treatment adherence tend to have healthier lifestyles, since they better comply with heart-healthy recommendations [8,9]. On the other hand, evidence indicates that adherence to drug therapy does not necessarily imply general healthy behaviour [9,10,11]. These conflicting findings suggest that an association between healthy lifestyle and good adherence to prescribed therapies is only found in individuals with certain profiles, the characterization of which may be beneficial for CVD management.

In addition to known benefits at the individual level, adherence to prescribed medication is associated with a long-term decrease in healthcare expenditure derived from hospital referrals, visits to primary care physicians and specialists, laboratory tests, and medication use [5,12,13]. On the other hand, evidence indicates that patients who begin and adhere to preventive treatments are more likely to seek out preventive health services, such as screening tests and vaccinations, than comparable patients who do not remain adherent [14]. This suggests that patients more engaged in practices consistent with a healthy lifestyle may account for greater healthcare utilization. Thus, further examination of the global impact of patient behavior, in terms of both lifestyle and persistence with medication, on healthcare utilization, results of interest.

Working population spend most of their life time in the workplace. This context likely influences their lifestyle (dietary habits, physical activity, smoking, or alcohol consumption) and also their adherence to prescribed therapies, especially in the case of shift works. The work setting represents, therefore, a unique opportunity to intervene with the objective of increasing the workers’ commitment with disease prevention.

In this study conducted in a workers’ cohort we sought to identify distinct profiles of new lipid-lowering drug users based on adherence to CVD prevention recommendations (healthy lifestyle and persistence with medication), and analyze differences between profiles in terms of comorbidities, co-treatments, and healthcare utilization.

## 2. Materials and Methods

### 2.1. Study Design and Participants

The Aragon Workers’ Health Study (AWHS) is a prospective longitudinal study aimed at evaluating the association between CVD risk factors and metabolic abnormalities and subclinical atherosclerosis in a middle-aged working population in Spain that was CVD-free at the beginning of the study. The cohort, recruited between February 2009 and December 2010, consists of 5678 workers at an automobile assembly plant in Zaragoza (Spain) who voluntarily agreed to participate in the study. Between 2011 and 2014, participants aged 39–59 years were invited to undergo more intensive monitoring consisting of subclinical atherosclerosis imaging, clinical and physical examination, and questionnaires on lifestyle factors. Further information on the AWHS can be found in the bibliography [15].

As shown in Figure 1, the present study included all AWHS participants in the more intensive monitoring group who began lipid-lowering therapy during the period 2010 to 2013 for primary CVD prevention. Given the low percentage of females in the AWHS, as well as recognized sex-related effects on health behaviors, the 23 female participants were excluded from our analyses. Individuals for which complete lifestyle data were unavailable were also excluded. Ultimately, 517 men were included in our analysis.

### 2.2. Data Sources and Variables

Drug prescription data were gathered from Farmasalud, the Medication Consumption Information System of Aragon. This database collects the following information on all prescriptions dispensed at pharmacies in Aragon via the public health care system (accounting for 98.5% of the population): anonymous patient code; sex; birth date; Anatomical Therapeutic Chemical (ATC) code of the prescribed drug; dispensing date; number of defined daily doses (DDD); and number of packages dispensed. Prescriptions issued by private physicians, insurance companies, and in-hospital consumption are not recorded in the Farmasalud database. All prescriptions for lipid-lowering drugs (ATC code C10 according to the World Health Organization ATC/DDD Index) issued between January 1, 2010 and December 31, 2014 were recorded. In addition, the number of CVD preventive co-treatments prescribed to each worker in the index year was determined by taking into account the existence of any prescription for antidiabetic (A10 code), antihypertensive (C02, C03, C07, C08, C09) or antithrombotic (B01) drugs. The drug burden for each worker was analyzed as the total number of pharmacological subgroups (i.e., different ATC codes at the 3rd level in the ATC classification system) prescribed during the index year. Each pharmacological subgroup includes drugs with a similar indication or mechanism of action, so that it constitutes a proxy of the burden of medication and, indirectly, of morbidity.

Clinical and laboratory data recorded by the physicians and nurses of the factory medical services at the annual workplace medical examination were collected for the period 2011 to 2014. The physical examination included measurement of height, weight, waist circumference, and blood pressure. It followed standardized protocols using validated procedures and instruments described in the standard operating procedures. The study conforms to the ISO9001-2008 quality standard. Waist circumference was measured with a GulicK II measuring tape model 67019. The highest point of the iliac crest and the lowest point of the costal margin were located in the mid-axillary line. The midpoint was calculated and the measuring tape was placed around the waist coinciding with that point and in a horizontal plane. Blood pressure was measured three consecutive times using an automatic oscillometric sphygmomanometer OMRON M10-IT (OMRON Healthcare Co. Ltd., Kyoto, Japan) with the participant sitting after a 5-min rest. The self-reported smoking status of the individuals in the preceding year was also recorded. Total cholesterol levels, fasting serum glucose, and glycated haemoglobin concentration were determined by enzyme analysis using the ILAB 650 analyzer (Instrumentation Laboratory; Bedford, MA, USA), and levels of Low-Density Lipoprotein (LDL)-cholesterol were estimated using the Friedewald equation [16]. The following cut-off points were applied based on current European Guidelines [2]: overweight, Body Mass Index (BMI) ≥25 and <30; obesity, BMI ≥ 30; hypertension, diastolic blood pressure ≥90 mmHg and systolic blood pressure ≥140 mmHg; hypercholesterolemia, total cholesterol ≥200 mg/dl and LDL-cholesterol ≥115 mg/dl; diabetes, fasting serum glucose ≥126 mg/dl and glycated hemoglobin ≥6.5%.

Participants also completed a questionnaire on sociodemographic characteristics including age, sex, education level, work shift, and type of work performed. The protocols for data collection are described in detail elsewhere [15].

Dietary habits were assessed by a face-to-face interview through a semi-quantitative food frequency questionnaire previously validated for the Spanish population [17]. The questionnaire consists of 136 questions on habitual food intake over the previous 12 months, and was administered by a trained dietician. Data derived from the questionnaire were subsequently converted into energy and nutrients using Spanish food composition tables [18,19]. Adherence to the Mediterranean diet was assessed using the Alternate Mediterranean Dietary (aMED) index [20]. The aMED score involves the following food groups and nutrients: fruit; vegetables; nuts; legumes; whole grains; fish; ratio of monounsaturated to saturated fat; red and processed meats; and alcohol. The total aMED score ranges from 0–9, with higher scores reflecting greater adherence to a Mediterranean diet. Accordingly, the aMED score obtained for each worker was classified as low (0–3), medium (4–6), or high (7–9).

Leisure time and physical activity were assessed using the validated Spanish version [21,22] of the Nurses’ Health Study [23] and the Health Professionals’ Follow-up study [24] physical activity questionnaires. Participants were asked about the frequency of and time devoted to different types of physical activity. This time was multiplied by the corresponding metabolic cost according to Ainsworth’s compendium for physical activities [25], and expressed in metabolic equivalents (MET). The total level of physical activity per week per individual was estimated by adding the METs for all activities. Sedentary time was calculated as the number of hours that each worker reported spending seated during a weekday, during both working and leisure time. Sleep time, taken as the usual number of hours of sleep per weekday, was also recorded for each individual.

Finally, information on healthcare utilization during the period 2015 to 2018 was gathered from different administrative health databases (i.e., primary, specialist, hospital, and emergency care databases) and linked with the rest of the data using the anonymized patient code. The number of visits to the emergency department, general practitioners, and specialist physicians, as well as the number of hospitalizations for each worker for the period 2015 to 2018 were recorded. In addition, the number of individual hospital discharges with a principal diagnosis code corresponding to CVD (G45, G46, G81-G83, I20-I28, I46, I49.0, I50, and I60-I79 in the International Classification of Diseases, Tenth Revision (CD-10)) was also recorded. Both the number and burden of chronic diseases for each worker in 2018 were retrieved from the Adjusted Morbidity Groups (GMA) database. GMA is a method developed to calculate an individual’s morbidity burden and adapted to the Spanish healthcare system [26]. Finally, the number and date of all-cause deaths, for the period 2015 to 2018, were obtained from the Spanish National Mortality Registry.

### 2.3. Analyses

For each individual, the index date was defined as the date of dispensing of the first lipid-lowering drug. Analyses were restricted to new users, defined as those who had not received any lipid-lowering drug prescription during the 6 months preceding the index date (i.e., between 1 July 2010 and 31 December 2013).

Persistence with medication was defined as the time from initiation until discontinuation during a follow-up period of 1 year (between 2010 and 2014). Applying criteria utilized in previous studies [27], the number of days’ supply for each lipid-lowering prescription was estimated based on the usual dosage and form of presentation of the drug. Statin and fibrate prescriptions were assigned a DDD of 28 and 30, respectively. A worker was classified as persistent if there was no gap between 2 refills exceeding 2.5 times the duration of the previous prescription during the 1-year follow-up period. The selection of this criterium was based on sensitivity analyses performed in a previous study [27]. The accumulation of supplies over time was not considered.

The baseline characteristics of the study population as well as the proportion of individuals considered persistent with lipid-lowering medication were recorded. Categorical variables were expressed as the frequency (n) and proportion (%), and continuous variables as the mean and standard deviation. The European cardiovascular score for countries with low CVD risk was also calculated for each individual [28].

We performed a cluster analysis to identify different aggregations or profiles of subjects according to their adherence to CVD prevention recommendations. Specifically, a 2-step cluster analysis was applied because it allows the use of both categorical and continuous variables. This technique automatically determines the optimal number of clusters needed to perform the data grouping based on the Bayesian Information Criterion (BIC). The log-likelihood distance was used to calculate the similitude between groups. Thus, individuals are assigned to groups based on maximum within-group similarity and between-group differences with respect to the variables initially considered of interest.

In the present study, these variables were as follows: adherence to the Mediterranean diet; energy intake; physical activity; sitting time; alcohol intake; smoking status; and persistence with therapy. The variables ultimately selected to define clusters must be independent from the rest of the variables conforming the clusters. The quality of fit of the resulting groups of clusters is evaluated using the silhouette measure of cohesion and separation, which contrasts the mean distance among elements within the same cluster with the average distance to elements in other clusters. A silhouette coefficient ≥0.2 is considered acceptable [29,30]. When a similar silhouette coefficient was obtained for different possible aggregations, the selection of the final group of clusters was based on the independence (no-correlation) and clinical coherence of the resulting variables.

Once the different profiles of workers were identified by cluster analysis, the corresponding data on sociodemographic characteristics, CVD risk factors, cardiovascular scores, lifestyle, burden of chronic disease, number of comorbidities and CVD preventive co-treatments, pharmacological subgroups, and contact with healthcare services were compared. An ANOVA test was applied for comparison of continuous variables with normal distribution (previously demonstrated by the Kolmogorov–Smirnov test), Kruskall–Wallis for no normal continuous variables and chi-square for categorical variables (Fisher test when expected frequencies were lower than 5). When there were more than 2 groups, we also performed pairwise comparisons adjusting for multiple testing (Tukey when row-variable was normal-distributed and Benjamini and Hochberg method otherwise).

Two-step cluster analyses were performed using SPSS version 22 (license University of Zaragoza), and all subsequent analyses were performed using R statistical software.

### 2.4. Ethical Aspects

All research was conducted in accordance with relevant guidelines, and study participants provided written informed consent to participate in the AWHS. All data used were anonymized, making patient identification impossible. The study was approved by the Aragon Research Ethics Committee (Project identification code PI07/09).

## 3. Results

The baseline characteristics of the study population are presented in Table 1. Among the 517 male workers, the mean age was 51.0 (SD 3.7) years and 48.3% had completed primary studies. Additionally, 78.7% performed rotating shift work and 85.1% performed manual work. Most were either current (38.1%) or former (44.3%) smokers, 59.6% were overweight, and 26.9% were obese. The distribution of other traditional CVD risk factors was as follows: hypertension, 17.7%; hypercholesterolemia, 83.3%; diabetes, 3.5%. Less than one-third of the study population was persistent with lipid-lowering therapy.

The four different clusters into which the 517 workers were aggregated were defined based on the following characteristics, in order of importance: adherence to the Mediterranean diet; persistence with lipid-lowering therapy; smoking status; and physical activity (Figure 2). For this group of clusters, the silhouette coefficient was 0.4, what indicates an acceptable quality of clustering. Cluster 1 (*n* = 113) comprised workers with healthier lifestyles (61.1% never smoked; 84.0% had medium-high adherence to the Mediterranean diet; physical activity was high, with a median energy expenditure of 34 METs-h/week), and 16.8% were persistent with therapy. By contrast, cluster 3 (*n* = 108) comprised workers with the least healthy habits (100.0% were current or former smokers, adherence to the Mediterranean diet was low, and median energy expenditure was low, at 28.4 METs-h/week). Moreover, all individuals in cluster 3 were non-persistent with therapy. Clusters 2 (*n* = 150) and 4 (*n* = 146) presented intermediate characteristics in relation to smoking, adherence to the Mediterranean diet, and physical activity. However, 100.0% of workers in cluster 2 were persistent with therapy, while none of the workers in cluster 4 were persistent.

Of the non-persistent individuals in clusters 1, 3, and 4, 17.7, 34.3, and 26.0%, respectively, corresponded to users who received one or several lipid-lowering drug prescription(s) at the index date but no additional prescriptions during the follow-up period.

The mean (SD) number of cigarettes per day was 12.81 (6.65), 15.15 (8.10), 15.08 (8.37), and 12.94 (7.99) for current smokers in clusters 1, 2, 3, and 4, respectively.

Comparison of other characteristics between clusters (Table 2) revealed differences in the type of work performed: sedentary work was performed by a larger proportion of workers in cluster 1 and by a smaller proportion of workers in clusters 2 and 3.

Overweight and obesity were the only traditional CVD risk factors for which significant differences were observed among clusters: cluster 2 contained a larger proportion (34.7%) of obese subjects, while the frequency of overweight was higher within clusters 3 and 4 (63.9 and 66.4%, respectively). Individuals within cluster 1 had a significantly lower cardiovascular score than those in the other clusters.

Comparison of healthcare utilization revealed that the number of visits to emergency services and hospitalizations was significantly lower in cluster 1 than in clusters 2 and 4, or in cluster 2, respectively. The mean number of visits to primary care for cluster 1 was lower compared with the observed in cluster 4, while the mean number of visits to specialized care was also lower in cluster 1 compared with cluster 3.

Workers within cluster 1 had a significantly lower number and burden of chronic diseases than those in cluster 3. No differences were observed between clusters in terms of the number of CVD preventive co-treatments or pharmacological subgroups.

Comparison of healthcare utilization revealed that the number of visits to emergency services, hospital, and primary and specialized care was significantly lower in cluster 1 than the other clusters.

## 4. Discussion

This observational study examined a population of new lipid-lowering drug users with low morbidity, albeit high levels of CVD risk factors, including hyperlipidaemia, smoking, and overweight and obesity, and overall poor persistence with lipid-lowering therapy. Within this population, we identified four different profiles or clusters according to adherence to CVD prevention recommendations. These clusters were defined based on the following variables: smoking status, diet, physical activity, and persistence with lipid-lowering therapy. Analysis of differences in sociodemographic, anthropometric, and clinical characteristics between the four clusters revealed particularly significant differences between clusters with the healthiest (cluster 1) and the least healthy (cluster 3) lifestyle behaviors. In cluster 1, the proportion of current or former smokers was lower, while adherence to the Mediterranean diet and daily practice of physical activity were higher. Moreover, in cluster 1 the daily energy intake, quality of the fats consumed, and alcohol intake were much more in line with medical recommendations than in cluster 3, in which these parameters were indicative of an unhealthy lifestyle.

In agreement with the healthy adherer effect, none of the individuals in cluster 3 were persistent with lipid-lowering therapy. However, contrary to expectations only 16.8% of those in cluster 1 showed good persistence, while all subjects in cluster 2, in which lifestyles were not particularly healthy, were persistent. The hypothesis of risk compensation offers one possible explanation for these latter findings. According to this hypothesis, individuals who undergo a risk-lowering intervention are more likely to engage in risky behaviors because they feel that their level of risk is effectively reduced by the medication [31]. Studies of users of other treatments for chronic diseases, including osteoporotic fracture patients treated with bisphosphonate [9] and adults treated with antiepileptic drugs [32], have also reported no evidence of a healthy adherer effect.

Our findings reveal the existence of different profiles among lipid-lowering drug users, who may adhere to therapy but not to healthy lifestyle behaviors necessary for prevention and treatment of CVD, and vice versa. Halava et al. [33] noted that the association between lifestyle factors and non-adherence to statin therapy varied according to patient CVD comorbidity status. In their study of a large cohort of public sector employees, individuals with CVD comorbidities who had several unhealthy lifestyle behaviors were at increased risk of non-adherence. Conversely, overweight, obesity, and former smoking were predictors of better adherence among those without CVD comorbidities. Given that our study population is relatively young, with low levels of morbidity, we were unable to investigate this association.

The highest cardiovascular score was observed for lipid-lowering drug users who did not follow medication recommendations and only partially followed lifestyle recommendations. This high risk is probably influenced by the high frequency of CVD risk factors other than hyperlipidaemia in our study population. Early detection of individuals with this profile would facilitate implementation of motivational education programs to improve lifestyle and medication-related behaviors and the ability to adopt and maintain good CVD self-management practices [6]. In this sense, interventions designed to increase the level of persistence should always take into account the multifactorial nature of this phenomenon [34]. Some possible explanations for non-persistence in our study cohort include the appearance of adverse effects associated with the new drug, as well as the asymptomatic nature of hyperlipidaemia, which results in a lack of awareness of the consequences of non-persistence. Our findings also point to a potential role of rotating shift work as a determinant of poor persistence, although no previous studies appear to have investigated this association. It has been widely demonstrated that a low educational level is predictive of poor long-term adherence [35,36]. In contrast, we found that the only cluster in which all subjects were persistent was the cluster with the highest proportion of workers who had only completed primary studies. These results were, however, not statistically significant. Furthermore, in cluster 1, in which the frequency of sedentary work (i.e., office work that requires a higher educational level) was higher, less than one fifth of subjects were persistent. While our results do not demonstrate an association between a higher educational level and better persistence with medication, they show that individuals with a high educational level (cluster 1) had an overall healthier lifestyle, as well as a lower frequency of overweight or obesity, a lower burden of morbidity, and a lower cardiovascular score than subjects in other clusters. Moreover, cluster 1 had a lower number of visits to emergency services, primary and specialized care, and fewer hospitalizations. Longer follow-up as our population ages will be required to confirm the differences observed between clusters. In their study of Spanish adults with a mean age at baseline of 68.6 years, Hernandez-Aceituno et al. [37] reported a statistically lower risk of polypharmacy and fewer visits to the primary care physician and hospitalizations in subjects with 5–6 healthy behaviors than in those with 0–1 healthy behaviors. The authors found no association between healthy behaviors and visits to medical specialists.

The literature regarding the association between adherence to medication and healthcare utilization and, consequently, healthcare expenditure, is contradictory. Simon-Tuval et al. [12] reported that initiation of therapy with CVD preventive drugs by adherent patients was followed by a decrease in total healthcare costs, mainly owing to a decrease in hospitalizations. In partially- and non-adherent individuals, the authors observed no significant changes in annual healthcare costs after beginning therapy. Conversely, in their study of a Spanish cohort of 1.7 million individuals who were prescribed a new medication in 2012, Aznar-Lou et al. [38] found that patients who were initially non-adherent to chronic medication showed a lower use of healthcare services than initially adherent patients. Brookhart et al. [14] examined the association between adherence to statin therapy and the use of preventive health services in a cohort of 20,783 new users of statins in Pennsylvania. After adjustment for age, sex, and various comorbid conditions, patients who adhered to therapy showed higher subsequent use of preventive health services such as prostate-specific antigen tests, faecal occult blood tests, screening mammograms, influenza vaccinations, and pneumococcal vaccinations. Costs associated with preventive tests requested by more compliant patients are unlikely to exceed, in the medium and long term, those associated with co-treatments, diagnostic tests, visits to a general practitioner, specialist physician, emergency department, or hospitalizations required when a patient presents a high CVD risk. Further research, comparing the true healthcare expenditure in both adherent and non-adherent patients will be of interest. In any case, maintaining good control of CVD risk factors from young ages is essential to avoid CVD morbimortality. One study of a longitudinal cohort of industrial employees from Chicago aged 18–74 found that favorable CVD health at younger ages increased survival by almost 4 years and delayed the onset of all-cause and CVD morbidity by 4.5 and 7 years, respectively, resulting in lower healthcare costs [39].

To the best of our knowledge, this is the first study to assess clusters of young healthy subjects classified according to their adherence to both medication and healthy lifestyle behaviors. The availability of individual, clinical and drug utilization data, as well as healthcare utilization data in populations of this size is uncommon, and provides a rare opportunity to examine numerous conditions and variables and obtain a snapshot of the different profiles of users of CVD preventive drugs. Further strengthening our findings, the data sources used contain validated data that have been previously analyzed in published studies [15,27].

Limitations of the present study should be borne in mind. First, both the unique profile of the study population according to its sociodemographic characteristics, as well as the voluntary sampling approach complicate extrapolation of results to general population. Nevertheless, extrapolation would be possible in the case of young and healthy men. In this regard, the observed levels of adherence to CVD prevention recommendations, especially to medication, appear in line with those reported in previous population studies. Second, the assessment of diet and physical activity consisted of personal interviews using questionnaires that, although validated, can lead to some misclassification of subjects, as occurs in most diet- and exercise-related studies. The low numbers of comorbidities, co-treatments, and healthcare utilization in the different clusters hindered the detection of statistically significant differences. These low numbers were largely due to the young age and good health status of the study population. Consequently, longer follow-up should be performed in future studies, which should also analyze in greater detail the evolution of persistence with lipid-lowering medication within each cluster and would likely reduce the presence of confounders that may have influence the results in health. For instance, it may be informative to differentiate between subjects based on the purpose of the visit (e.g., assistance or prevention). Additionally, information on the statin dosing regimen prescribed, the provision of advice by health professionals other than physicians or the family’s habits and lifestyle would be of interest to better interpret the results found. Finally, a common limitation of analyses of drug-dispensing data is the assumption that the purchase of a drug equates to its consumption. We assessed drug-taking behavior by analyzing persistence with therapy, which is considered a more appropriate indicator than adherence, as it takes into account the continuity of the patient’s purchasing of medication [27]. Nonetheless, development of a standard method for assessing persistence with medication is needed.

## 5. Conclusions

Individual control of CVD risk factors requires adherence to heart-healthy recommendations relating to both lifestyle and persistence with medication. In our young, active, and healthy population, in which lipid-lowering therapy was initiated for primary prevention of CVD, we observed four distinct profiles or aggregations. The “healthy adherer effect” was only observed in some of these profiles.

These findings raise doubts as to the appropriateness of recommending a single set of actions for primary prevention of CVD in healthy population. The influence of educational level on adherence to both medication and a healthy lifestyle should be further analyzed in depth.

A wealth of interventions to improve the employees’ health are usually implemented at the workplace. However, further efforts are needed to determine how to provide effective and adjusted advice that promotes behavioral change and allocate the necessary resources according to the worker’s profile or needs. This would facilitate the creation of healthy environments and, consequently, the achievement of better health outcomes in both healthy and high-risk populations.

## Figures and Tables

**Figure 1 nutrients-13-00723-f001:**
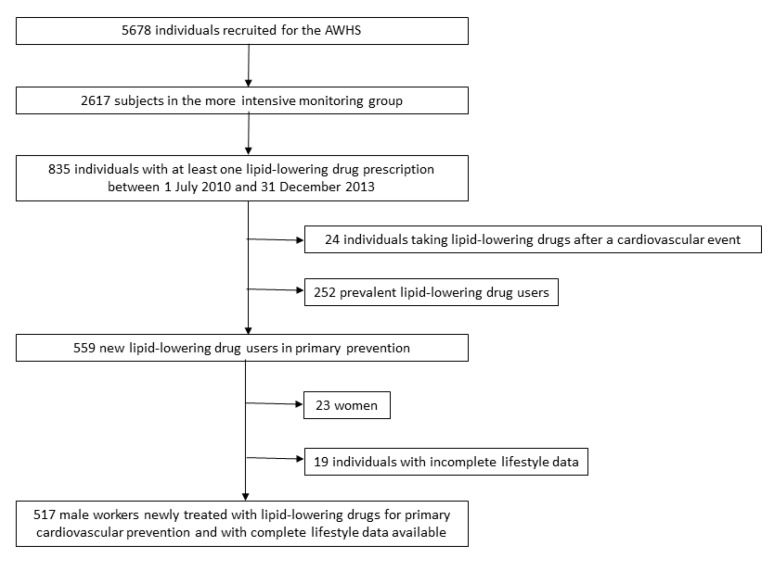
Flowchart of the study population.

**Figure 2 nutrients-13-00723-f002:**
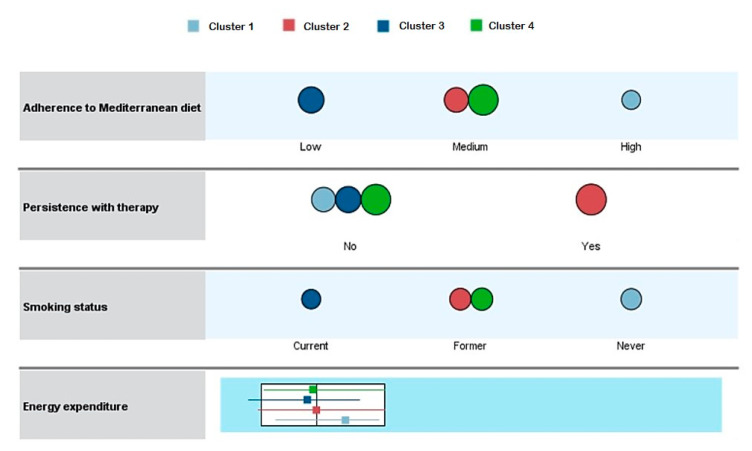
Comparison of the defining variables across the four established clusters.

**Table 1 nutrients-13-00723-t001:** Baseline characteristics of the study population.

Baseline Characteristics	Population (*N* = 517)
*Sociodemographic*	
Age, years	51.0 (3.7)
<50	103 (20.0%)
50–54	207 (40.0%)
>54	207 (40.0%)
Educational level	
Primary studies	250 (48.3%)
Secondary studies	66 (12.8%)
University studies + professional training	199 (38.5%)
Unknown	2 (0.4%)
Work shift	
Rotating shift	406 (78.7%)
Day shift	52 (10.1%)
Night shift	58 (11.2%)
Work type	
Manual	439 (85.1%)
Sedentary	77 (14.9%)
*Traditional CVD risk factors*	
Smoking	
Never	91 (17.6%)
Current	197 (38.1%)
Former	229 (44.3%)
BMI, kg/m^2^	28.2 (3.4)
Waist circumference, cm	98.3 (9.1)
Systolic blood pressure, mmHg	128.9 (15.2)
Diastolic blood pressure, mmHg	85.8 (9.53)
Total cholesterol, mg/dL	238.5 (36.4)
Glucose, mg/dL	101.0 (21.9)

N, number; CVD, cardiovascular disease; BMI, body mass index. Data are expressed as the mean (standard deviation) or number (%).

**Table 2 nutrients-13-00723-t002:** Comparison of the characteristics of the four clusters.

	Cluster 1 (*n* = 113)	Cluster 2 (*n* = 150)	Cluster 3 (*n* = 108)	Cluster 4 (*n* = 146)	*p*-Value
*Sociodemographic*					
Age, years					0.229
<50	29 (25.7%)	22 (14.7%)	24 (22.2%)	28 (19.2%)	
50–54	36 (31.8%)	65 (43.3%)	47 (43.5%)	59 (40.4%)	
>54	48 (42.5%)	63 (42.0%)	37 (34.3%)	59 (40.4%)	
Educational level					0.484
Primary studies	50 (45.0%)	81 (54.0%)	50 (46.3%)	69 (47.3%)	
Secondary studies	14 (12.6%)	14 (9.3%)	19 (17.6%)	19 (13.0%)	
University studies + professional training	47 (42.3%)	55 (36.7%)	39 (36.1%)	58 (39.7%)	
Work shift					0.179
Rotation shift	83 (74.1%)	121 (80.7%)	92 (85.2%)	110 (75.3%)	
Day shift	17 (15.2%)	12 (8.0%)	5 (4.6%)	18 (12.3%)	
Night shift	12 (10.7%)	17 (11.3%)	11 (10.2%)	18 (12.3%)	
Work type					0.001 *
Manual	86 (76.8%)	135 (90.0%)	98 (90.7%)	120 (82.2%)	
Sedentary	26 (23.2%)	15 (10.0%)	10 (9.3%)	26 (17.8%)	
*Traditional CVD risk factors*					
BMI, kg/m^2^					0.049 *
Normal weight (BMI < 25)	21 (18.6%)	21 (14.0%)	15 (13.9%)	13 (8.9%)	
Overweight (25 ≤ BMI < 30)	65 (57.5%)	77 (51.3%)	69 (63.9%)	97 (66.4%)	
Obesity (BMI ≥ 30)	27 (23.9%)	52 (34.7%)	24 (22.2%)	36 (24.7%)	
Waist circumference, cm	97.6 (9.7)	99.0 (8.8)	97.1 (8.0)	99.0 (9.6)	>0.1
Hypertension					0.361
Yes	18 (15.9%)	25 (16.9%)	25 (23.6%)	23 (15.8%)	
No	95 (84.1%)	123 (83.1%)	81 (76.4%)	123 (84.2%)	
Hypercholesterolemia					0.143
Yes	91 (82.0%)	124 (87.9%)	76 (76.8%)	117 (84.2%)	
No	20 (18.0%)	17 (12.1%)	23 (23.4%)	22 (15.8%)	
Diabetes					0.377
Yes	7 (6.2%)	4 (2.7%)	2 (1.9%)	5 (3.5%)	
No	106 (93.8%)	146 (97.3%)	104 (98.1%)	139 (96.5%)	
*Cardiovascular score*	1.6 (1.0) ^b,c,d^	1.9 (1.2) ^a^	2.0 (1.3) ^a^	2.1 (1.5) ^a^	0.026 *
*Lifestyle*					
Alcohol intake, g/day					0.032 *
<40	3 (2.7%)	8 (5.3%)	10 (9.3%)	7 (4.8%)	
41–60	103 (91.2%)	124 (82.7%)	78 (72.2%)	122 (83.6%)	
>60	7 (6.2%)	18 (12.0%)	20 (18.5%)	17 (11.6%)	
Energy intake, kcal/day	2745.2 (674.1) ^b,c^	2902.2 (709.5) ^a^	2932.8 (707.9) ^a^	2902.7 (734.5)	0.093
Carbohydrates, %	44.3 (6.0)	45.5 (6.6)	44.6 (7.3)	44.4 (6.6)	0.376
Proteins, %	15.3 (2.0) ^b,c^	14.7 (2.1) ^a,d^	14.8 (2.3) ^a,d^	15.5 (2.5) ^b,c^	0.009 *
Fats, %	35.4 (4.7)	33.9 (5.3)	34.8 (6.2)	34.8 (5.3)	0.181
Ratio monounsaturated: saturated fatty acids					0.001 *
1st tertile	27 (23.9%)	50 (33.3%)	51 (47.2%)	44 (30.1%)	
2nd tertile	34 (30.1%)	57 (38.0%)	32 (29.6%)	51 (34.9%)	
3rd tertile	52 (46.0%)	43 (28.7%)	25 (23.1%)	51 (34.9%)	
Sleeping time, h/weekday	6.3 (0.9)	6.3 (1.1)	6.3 (0.9)	6.1 (1.0)	0.015 *
Sitting time, h/weekday	7.9 (2.4)	8.1 (2.3)	8.3 (2.3)	8.1 (2.3)	0.818
*Healthcare utilization*					
Number of chronic diseases	3.0 (2.0) ^c^	3.2 (1.6)	3.5 (1.8) ^a^	3.3 (1.9)	0.107
Burden of chronic disease	4.2 (2.8) ^c^	4.9 (3.0)	5.2 (3.0) ^a^	4.9 (3.0)	0.111
Number of CVD preventive co-treatments					0.578
0–1	99 (87.6%)	135 (90.0%)	99 (91.7%)	135 (92.5%)	
>1	14 (12.4%)	15 (10.0%)	9 (8.3%)	11 (7.5%)	
Number of pharmacological subgroups	1.8 (1.2)	1.7 (1.0)	1.7 (1.1)	1.7 (1.0)	0.649
Visits to emergency services	0.7 (1.3) ^b,d^	1.0 (1.4) ^a^	1.0 (1.8)	1.1 (1.4)^a^	0.048
Number of hospitalizations	0.5 (1.3) ^b^	0.6 (1.0) ^a^	0.4 (0.9)	0.5 (1.0)	0.076
Number of hospitalizations due to CVD	0.0 (0.2)	0.1 (0.4)	0.0 (0.2)	0.1 (0.3)	0.169
Number of visits to primary care	28.4 (27.1) ^d^	32.6 (27.3)	31.7 (31.5)	39.0 (36.2) ^a^	0.061
Number of visits to specialized care	9.6 (12.1) ^c^	12.1 (13.8)	12.6 (13.4) ^a^	10.6 (12.0)	0.1
Number of visits to specialized cardiology care	0.6 (1.8)	0.8 (1.9)	0.9 (3.9)	0.8 (1.9)	0.4
Number of all-cause deaths	0	3	0	1	

CVD, cardiovascular disease; BMI, body mass index. Data are expressed as the mean (standard deviation) or number (%). Healthcare utilization corresponds to the period 2015–2018. The P-value corresponds to differences between the four clusters as determined by analysis of variance (ANOVA), Kruskal-Wallis, or chi-squared tests as appropriate. Superscripts a, b, c and d indicate significant differences between two specific clusters: **a**, vs. cluster 1; **b**, vs. cluster 2; **c**, vs. cluster 3; **d,** vs. cluster 4. In all cases, differences were considered statistically significant at *p* < 0.05 and represented as *.

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
