# Peer review of "Identifying Clusters of Adherence to Cardiovascular Risk Reduction Behaviors and Persistence with Medication in New Lipid-Lowering Drug Users. Impact on Healthcare Utilization"

_nutrients, 2021, doi:10.3390/nu13030723_

Round 1
Reviewer 1 Report
In their original research article, Sara Malo and colleagues develop a comprehensive data evaluation of a cohort comprising 517 males in terms of their sociodemographic characteristics, dietary habits, leisure time and physical activity habits and healthcare utilization with the purpose of identifying profiles of specific lipid-lowering drug users with adherence to a healthy lifestyle and persistence with medication.
The work is well written, and the contextualization done in the introduction is adequate. Though it has not been possible to identify a typical profile with good adherence to healthy behaviours and persistence with medication, it is my understanding that the Authors addressed all the essential points and raised important issues in the discussion.
Prior to publishing however, the Authors should address some minor details.
Specific comments:
1) Figure 1: Information in the flow chart is hard to read due to letter size/definition. I would ask the Authors to prepare a clearer scheme.
2) P.4, line 130: “BMI” – The Authors should stat what BMI stands for the first time it is mentioned in the text.
3) P.4, lines 130 – 134: I understand that the cut-off points used are quite standard, however since, for example, according to the population under study different cut-off points may be suggested to diagnose hypercholesterolemia (Cutoff Point Separating Affected and Unaffected Familial Hypercholesterolemic Patients Validated by LDL-receptor Gene Mutants, Journal of Atherosclerosis and Thrombosis, 12, 2005), I would ask the Authors to support the choice of the cut-off points they used with suitable references or mention to European Guidelines
4) Table 2: “Sleeping time, h/week” and “Sitting time, h/week” – I believe it should be “h/weekday”.
5) P. 10, line 355: “the highest proportion of workers who had completed” – I would suggest adding “only”: “the highest proportion of workers who had only completed”
6) P. 10, line 372: “There” – “The”
Author Response
First, we would like to thank the reviewer for the provided comments, which have been very useful to improve this last version of the paper. The modifications made appear highlighted using the "Track Changes" function in the marked-up copy of the manuscript. Next, we proceed to answer in detail all the points addressed in the received letter:
Specific comments:
1) Figure 1: Information in the flow chart is hard to read due to letter size/definition. I would ask the Authors to prepare a clearer scheme.
Answer from authors:
As requested, the flow chart has been redesigned by increasing the font size. We hope that it is clearer to read now.
2) P.4, line 130: “BMI” – The Authors should stat what BMI stands for the first time it is mentioned in the text.
Answer from authors:
The acronyms BMI have been defined in line 143 (Body Mass Index (BMI)).
3) P.4, lines 130 – 134: I understand that the cut-off points used are quite standard, however since, for example, according to the population under study different cut-off points may be suggested to diagnose hypercholesterolemia (Cutoff Point Separating Affected and Unaffected Familial Hypercholesterolemic Patients Validated by LDL-receptor Gene Mutants, Journal of Atherosclerosis and Thrombosis, 12, 2005), I would ask the Authors to support the choice of the cut-off points they used with suitable references or mention to European Guidelines
Answer from authors:
These cut-off points were applied based on current European Guidelines (Piepoli MF, et al. 2016 European Guidelines on cardiovascular disease prevention in clinical practice. Eur Heart J 2016;37:2315-2381), so this information has been incorporated in line 142: “The following cut-off points were applied based on current European Guidelines [2]:”
4) Table 2: “Sleeping time, h/week” and “Sitting time, h/week” – I believe it should be “h/weekday”.
Answer from authors:
We agree with the reviewer. The two typos have been corrected in Table 2.
5) P. 10, line 355: “the highest proportion of workers who had completed” – I would suggest adding “only”: “the highest proportion of workers who had only completed”
Answer from authors:
The sentence has been corrected as indicated.
6) P. 10, line 372: “There” – “The”
Answer from authors:
The typo has been corrected. Thank you.
Reviewer 2 Report
Thank you for the opportunity to read this manuscript (ID nutrients-1116093).It is a good study.However, I have a few questions and comments that I will post below.
ABSTRACT section
I have no comments.
INTRO section
I have no comments.
METHODS section
Who did the medical tests?
What tools were used for anthropometric measurements? What method was used to measure the waist circumference (e.g. in people with large abdominal obesity)? What cut-off point was adopted for the WC? Please provide the name of the blood pressure measuring device. Was the blood pressure measured once or twice and the mean calculated? If twice, what was the time interval between measurements? Did the participant complete the questionnaires independently or with the participation of the interviewer? What ingredients are included in the Mediterranean diet? Physical activity in leisure was measured, inter alia, using the Nurses' Health Study questionnaire. Is this the right tool to measure PA in men? If so, please explain why. As suggested by the administrator, please quote the address https://www.nurseshealthstudy.org/ and put it on the list of references.
ANALYSES subsection
Lines: 199-201. Rather, the automatic technique was chosen because it could be manual. What was the adopted measure of distance between objects and what method of linking objects? What post hoc test was used for multiple comparisons (ANOVA, Kruskal-Wallis test)? Exactly, ANOVA. The authors write about this test in the explanation below table 2 (line 287) but do not mention in the ANALYSES subsection. Which test was used for the comparisons of groups k> 2? If also ANOVA, what test was the homogeneity of variance tested with?
RESULTS section
In the text of this section, the authors repeat the information contained in the tables. I advise against. Also in the text of this section, the exact values of P were repeated on the basis of the tables, but only if they were significant. I wouldn't demonize P's values. It's a good overall thing. It should be used (or perhaps it should be) because it is statistical information, but is it also scientific?
DISCUSSION section
Lines 331-337.
In my opinion, this paragraph is a bit out of this fairy tale.
CONCLUSIONS section
Lines 434-441. These are rather results than conclusions.
REFERENCES section
As I mentioned in the METHODS section, as recommended by the authors of the questionnaire, please consider adding https://www.nurseshealthstudy.org/.
Author Response
First, we would like to thank the reviewer for the provided comments, which have been very useful to improve this last version of the paper. The modifications made appear highlighted using the "Track Changes" function in the marked-up copy of the manuscript. Next, we proceed to answer in detail all the points addressed in the received letter:
METHODS section
Who did the medical tests?
This information has been included in the Data sources and variables subsection (lines 124-125):
“(…) Clinical and laboratory data recorded by the physicians and nurses of the factory medical services at the annual workplace medical examination were collected for the period 2011–2014. (…)”
What tools were used for anthropometric measurements?
In lines 127-130 it has been indicated that: “The physical examination included measurement of height, weight, waist circumference, and blood pressure. It followed standardized protocols using validated procedures and instruments described in the standard operating procedures. The study conforms to the ISO9001-2008 quality standard. (…)“
What method was used to measure the waist circumference (e.g. in people with large abdominal obesity)? What cut-off point was adopted for the WC?
The measurement of waist circumference has been explained in lines 130-134: “(…) Waist circumference was measured with a GulicK II measuring tape model 67019. The highest point of the iliac crest and the lowest point of the costal margin were located in the mid-axillary line. The midpoint was calculated and the measuring tape was placed around the waist coinciding with that point and in a horizontal plane. (…)”
With regards to the second part of the question, we would like to clarify that any cut-off point was considered for the waist circumference. In the Results section, only the mean (SD) of this anthropometric measure in both the study population and the four clusters is presented, without being interpreted nor categorized.
Please provide the name of the blood pressure measuring device. Was the blood pressure measured once or twice and the mean calculated? If twice, what was the time interval between measurements?
This information has been included in lines 134-136: “(…) Blood pressure was measured three consecutive times using an automatic oscillometric sphygmomanometer OMRON M10-IT (OMRON Healthcare Co. Ltd., Japan) with the participant sitting after a 5-min rest. (…)”
Did the participant complete the questionnaires independently or with the participation of the interviewer?
It has been clarified in line 150: “Dietary habits were assessed by a face-to-face interview through a by means of a semi-quantitative food frequency questionnaire previously validated for the Spanish population [17]. “
What ingredients are included in the Mediterranean diet?
This information has been included in lines 157-159: “(…) The aMED score involves the following food groups and nutrients: fruit; vegetables; nuts; legumes; whole grains; fish; ratio of monounsaturated to saturated fat; red and processed meats; and alcohol. (…)”
Physical activity in leisure was measured, inter alia, using the Nurses' Health Study questionnaire. Is this the right tool to measure PA in men? If so, please explain why. As suggested by the administrator, please quote the address https://www.nurseshealthstudy.org/ and put it on the list of references.
Answer from authors:
Physical activity in leisure was measured by a questionnaire designed from both the Nurses’ Health Study and the Health Professionals’ Follow-up study questionnaires. The first one was only based in women but the second one included both men and women. Moreover, the questionnaire was validated in the Spanish population and has been utilized in several cohort studies (e.g. SUN cohort (see reference 21)). This information and the corresponding references are included in lines 163-165:
“Leisure time and physical activity were assessed using the validated Spanish version [21,22] of the Nurses’ Health Study [2223] and the Health Professionals’ Follow-up study [2324] physical activity questionnaires.”
The reference corresponding to the Nurses' Health Study has been included (reference number 23).
ANALYSES subsection
Lines: 199-201. Rather, the automatic technique was chosen because it could be manual. What was the adopted measure of distance between objects and what method of linking objects?
What post hoc test was used for multiple comparisons (ANOVA, Kruskal-Wallis test)? Exactly, ANOVA. The authors write about this test in the explanation below table 2 (line 287) but do not mention in the ANALYSES subsection.
Which test was used for the comparisons of groups k> 2? If also ANOVA, what test was the homogeneity of variance tested with?
Answer from authors:
The first question related to the measure of distance adopted and the method of linking objects is answered in the following sentence (lines 214-219): “This technique automatically determines the optimal number of clusters needed to perform the data grouping based on the Bayesian Information Criterion (BIC). The log-likelihood distance was used to calculate the similitude between groups. AlsoThus, it assigns individuals are assigned to groups based on maximum within-group similarity and between-group differences with respect to the variables initially considered of interest. (…)”
As requested, the tests used in post hoc analyses have also been indicated (lines 235-242): “(…) ANOVA test was applied for comparison of continuous variables with normal distribution (previously demonstrated by the Kolmogorov-Smirnov test), Kruskal-Wallis for no normal continuous variables and chi-square for categorical variables (Fisher test when expected frequencies were lower than 5). using Kruskal-Wallis or chi-squared tests, as appropriate. When there were more than 2 groups, we also performed pairwise comparisons adjusting for multiple testing (Tukey when row-variable was normal-distributed and Benjamini & Hochberg method otherwise).”
RESULTS section
In the text of this section, the authors repeat the information contained in the tables. I advise against. Also in the text of this section, the exact values of P were repeated on the basis of the tables, but only if they were significant. I wouldn't demonize P's values. It's a good overall thing. It should be used (or perhaps it should be) because it is statistical information, but is it also scientific?
Answer from authors:
As requested, some of the information included in both the text and the tables has been removed from the text. Moreover, the exact P-values have been deleted from the text.
Line 259:
“Less than one-third (32.7%) of the study population was persistent with lipid-lowering therapy.”
Lines 294-299:
Overweight/obesity was the only traditional CVD risk factor for which significant differences were observed among clusters (p=0.049): cluster 2 contained a larger proportion (34.7%) of obese subjects, while the frequency of overweight was higher within clusters 3 and 4 (63.9% and 66.4%, respectively). Individuals within cluster 1 had a significantly lower cardiovascular score than those in the other clusters (p=0.026).
Lines 302-305:
“The mean number of visits to primary care for cluster 1 was 28.4lower compared with 39.0the observed in cluster 4 (P<0.05), while the mean number of visits to specialized care was 9.6also lower in cluster 1 compared with 12.6 in cluster 3 (P<0.05).”
Lines 320-322:
“Specifically, the mean number of visits to primary care for cluster 1 was 28.4 compared with 39.0 in cluster 4, while the mean number of visits to specialized care was 9.6 compared with 12.6 in cluster 3.”
DISCUSSION section
Lines 331-337. In my opinion, this paragraph is a bit out of this fairy tale.
Answer from authors:
We have moved this paragraph to lines 342-349 in order to better contextualize the idea. Now, it is presented as follows:
“(…) In agreement with the healthy adherer effect, none of the individuals in cluster 3 were persistent with lipid-lowering therapy. However, contrary to expectations only 16.8% of those in cluster 1 showed good persistence, while all subjects in cluster 2, in which lifestyles were not particularly healthy, were persistent. The hypothesis of risk compensation offers one possible explanation for these latter findings. According to this hypothesis, individuals who undergo a risk-lowering intervention are more likely to engage in risky behaviours because they feel that their level of risk is effectively reduced by the medication [321]. Studies of users of other treatments for chronic diseases, including osteoporotic fracture patients treated with bisphosphonate [9] and adults treated with antiepileptic drugs [32], have also reported no evidence of a healthy adherer effect. (…)”
CONCLUSIONS section
Lines 434-441. These are rather results than conclusions.
Answer from authors:
The paragraph indicated has been shortened and it now reads:
“Individual control of CVD risk factors requires adherence to heart-healthy recommendations relating to both lifestyle and persistence with medication. In our young, active and healthy population, in which lipid-lowering therapy was initiated for primary prevention of CVD, we observed four distinct profiles or aggregations. Two corresponded to the healthiest and the least healthy, respectively, while the other two presented intermediate characteristics, and differed from one another in terms of the proportion of subjects that were persistent with therapy (100% versus 0%). Thus, tThe “healthy adherer effect” was only observed in some of these profilescertain profiles of the subjects studied.
REFERENCES section
As I mentioned in the METHODS section, as recommended by the authors of the questionnaire, please consider adding https://www.nurseshealthstudy.org/.
Answer from authors:
This reference has been added to the list (reference number 23).
Reviewer 3 Report
An interesting paper regarding compliance to CVD prevention interventions. Did the authors found any difference in compliance between patients receving statin early in the morning and those receiving statin at night before sleep?Also, were any patients advised by other health workers (except from physicians, eg dietologists ) and how this affected adherence. Did spouse's or family's lifestyle affected patients' adherence/lifestyle?
Author Response
First, we would like to thank the reviewer for the provided comments, which have been very useful to improve this last version of the paper. The modifications made appear highlighted using the "Track Changes" function in the marked-up copy of the manuscript. Next, we proceed to answer in detail points addressed in the received letter:
An interesting paper regarding compliance to CVD prevention interventions. Did the authors found any difference in compliance between patients receving statin early in the morning and those receiving statin at night before sleep? Also, were any patients advised by other health workers (except from physicians, eg dietologists ) and how this affected adherence. Did spouse's or family's lifestyle affected patients' adherence/lifestyle?
Answer from authors:
We coincide with the reviewer that this information would be of interest but unfortunately was not available. We have explained it in the Discussion section, lines 455-459:
“(…) For instance, Iit may also be informative to differentiate between subjects based on the purpose of the visit (e.g. assistance or prevention). Additionally, information on the statin dosing regimen prescribed, the provision of advice by health professionals other than physicians or the family's habits and lifestyle would be of interest to better interpret the results found. (…)”